**OUJ-FTC-6**
**OCHA-PP-367**

# Anomalous diffusion in a randomly modulated velocity field

Noriaki Aibara[1], Naoaki Fujimoto[2], So Katagiri[1], Yutaka Matsuo[3], Yoshiki Matsuoka[1†],
Akio Sugamoto[4], Ken Yokoyama[1], and Tsukasa Yumibayashi[5]

[1]*Nature and Environment, Faculty of Liberal Arts, The Open University of Japan, Chiba 261-8586, Japan*

[2]*Department of Information Design, Faculty of Art and Design, Tama Art University, Hachioji, 192-0394 Japan*

[3]*Department of Physics, Trans-scale Quantum Science Institute, Mathematics and Informatics Center, University of Tokyo, Hongo 7-3-1, Bunkyo-ku, Tokyo 113-0033, Japan*

[4]*Department of Physics, Graduate School of Humanities and Sciences, Ochanomizu University, 2-1-1 Otsuka, Bunkyo-ku, Tokyo 112-8610, Japan*

[5]*BrainPad Inc., 3-2-10 Shirokanedai, Minato-ku, Tokyo 108-0071, Japan*

[†]Email: machia1805@gmail.com

## Abstract

This paper proposes a simple model of anomalous diffusion, in which a particle moves with the velocity field induced by a single "dipole" (a doublet or a pair of source and sink), whose moment is modulated randomly at each time step. A motivation to introduce such a model is that it may serve as a toy model to investigate an anomalous diffusion of fluid particles in turbulence. We perform a numerical simulation of the fractal dimension of the trajectory using periodic boundary conditions in two and three dimensions. For a wide range of the dipole moment, we estimate the fractal dimension of the trajectory to be 1.5–1.9 (2D) and 1.6–2.7 (3D).

# 1    Introduction

Brownian motion has a long history since Einstein's celebrated work. In recent years [1], anomalous diffusion has attracted attention, where the diffusion rate is different from that of a usual random walk. It has been observed not only in physical phenomena [2, 3, 4] but also in various places such as economy [5, 6], the animal group behavior [7, 8], and diffusion phenomena in biological organisms [9, 10].

A phenomenon known as Lévy Flight characterizes anomalous diffusion [11]. It is a sequence of particles' motion, with a random combination of local moves and hopping to a distant location. One observes a similar move in the motion of fluid particles [12, 13, 14, 15].

This paper proposes a toy model of fluid particles with such behavior. It consists of a particle that moves with the velocity field caused by a "dipole" with a randomly modulated moment located at the origin. As long as the particle moves away from the dipole location, it is the usual Brownian motion. However, when a particle approaches the dipole, it is bounced off by the singularity of the velocity field. If we impose the periodic boundary condition, it flies to another site. This movement creates Lévy Flight and fractals in the trajectory of the particle. We investigate the fractal dimension of this dipole model using a box-counting method to numerically evaluate the trajectories of particles in two and three dimensions. We expect that this model will be helpful to understand turbulent phenomena as well as the passive scalar field theory of turbulence [16, 17, 18]. One may identify the random dipole as a simplification of the vortex filament whose shape and orientation change randomly in turbulence. One may identify the velocity field with the fluid motion including the vortex filaments.

Our hydrodynamical model with dipole modulation is triggered by a mathematical work on Riemann's mapping theorem [19], which studied a successive application of conformal mappings to a complex domain $\mathbb{D}$, the mapping $z \to f(z, t)$ ($z \in \mathbb{C}$) at each time step $t$. The conformal mappings follow the so-called "Schramm Löwner evolution (SLE)" equation:

$$\frac{\partial}{\partial t} f(z, t) = -z \frac{z + \zeta(t)}{z - \zeta(t)} \frac{\partial}{\partial z} f(z, t). \tag{1}$$

It implies the following differential equation for the boundary curve $\partial \mathbb{D} = \{z(t)\}$,

$$\frac{dz(t)}{dt} = -z \frac{z + \zeta(t)}{z - \zeta(t)}. \tag{2}$$

Here, there exists a very interesting fact that the boundary curve becomes "fractal", having fractal dimension $D_f$, if the angle of $\zeta(t)$ is randomly modulated while its magnitude being fixed to a constant $\sqrt{\kappa}$ [20, 21]. $D_f$ is determined by $\sqrt{\kappa}$. The fractal dimensions $D_f$ estimated in SLE, were compared with the simulations of turbulence in 2D (dimensional) hydrodynamics. Although the SLE model may reproduce a fractal dimension $D_f = 4/3$, which is predicted by applying the scaling hypothesis of Kolmogorov and Kraichnan [22, 23, 24, 25, 26] to some curve which envelopes the cluster of vortices [27], it is by no means clear why SLE is relevant to the turbulence in hydrodynamics. Under these circumstances, we are going to construct a toy model, in which the similar phenomenon of fractals in SLE is expected to occur for a real stream line in hydrodynamics, when the fluid velocity is strongly fluctuated so as to generate the turbulence. Our model so obtained will be given in the next section. The model is directly related to the hydrodynamics in any spacial

dimensions $D = 2, 3, \cdots$, in which the emission of "eddies" in turbulence is simplified, in our toy model, to the emission of a "dipole" composed of a pair of source and sink of the fluid. Random production of "eddies" is incorporated as random modulation about the direction of the dipole moment. Therefore, our model can be viewed as that the SLE in mathematics is reformed so that it may adapt to the hydrodynamics in physics. In other words, the introduction of dipole with modulation, can be the field theoretical pair creation of fluid particles, or the string theoretical creation of a vortex ring. The dynamics of these pair creations may sow the seeds of the turbulence in hydrodynamics, but we need further investigations, before establishing it. Also, there are also literature discussing the relationship between turbulence and SLE using random fields [28, 29, 30], and it is expected that there may be a correspondence between these and our model.

We organize the paper as follows. In the next section, we define our model. The spatial dimension is arbitrary. Section 3 gives the main result of the simulation of the particle trajectories in 2D and 3D and of the estimate of the fractal dimension for various parameters, including the dipole moment. Section 4 gives more detail of the simulation, focusing on the dependence of various parameters. Section 5 summarizes the results of this paper. In the last section, we propose some theoretical ideas which may be helpful in the future.

**Note added:** After we submitted the first version of this paper, we realized that Lévy flight induced by random dipoles was also discussed in a recent paper [31] with a different setup. We thank Kiyoshi Kanazawa for pointing it out to us.

## 2 Particle motion in a randomly modulated velocity field

We consider a motion of a particle in $D$ dimensions,

$$\frac{d\boldsymbol{x}(t)}{dt} = \boldsymbol{V}_\zeta(\boldsymbol{x}(t); t), \quad \nabla \boldsymbol{V}_\zeta(\boldsymbol{x}; t) = \sum_i Q_i \delta^{(D)}(\boldsymbol{x} - \boldsymbol{\zeta}_i(t)) \tag{3}$$

where $\boldsymbol{\zeta}_i \in \mathbb{R}^D$ $(i = 1, \cdots, m)$ are the location of the source and $Q_i$ is the charge at $\boldsymbol{\zeta}_i$. $\boldsymbol{x} \in \mathbb{R}^D$ is the coordinate of the particle. In terms of Green's function of the Laplacian, one may write,

$$\boldsymbol{V}_\zeta(\boldsymbol{x}) = \sum_i Q_i \nabla G(\boldsymbol{x} - \boldsymbol{\zeta}_i(t)), \quad \Delta G(\boldsymbol{x} - \boldsymbol{y}) = \delta^{(D)}(\boldsymbol{x} - \boldsymbol{y}). \tag{4}$$

In $D$ dimensions, Green's function is

$$G(\boldsymbol{x}) = \begin{cases} -((D-2)S_{D-1}r^{D-2})^{-1} & D \geq 3 \\ \frac{1}{2\pi} \ln r & D = 2 \end{cases} \tag{5}$$

with $r = |\boldsymbol{x}|$ and $S_{D-1} = 2\pi^{D/2}/\Gamma(D/2)$.

In this paper, we treat $\boldsymbol{\zeta_i} \in \mathbb{R}^D$ as the random variables, which change at each time step, $\{\boldsymbol{\zeta}_i(t)\} = \{\boldsymbol{\zeta}_i(t_0) \to \boldsymbol{\zeta}_i(t_1) \to \boldsymbol{\zeta}_i(t_2) \to \cdots\}$, which makes eq.(3) a stochastic differential equation. We will study the trajectory of the particle $\{\boldsymbol{x}\}$ and evaluate the fractal dimension.

A motivation to study such a model is finding a toy model that captures some aspects of fluid turbulence. We note that one may describe the Euler equation of the fluid dynamics

in terms of vortex filaments, where each vortex moves as the potential flow with the other vortices as sources. In turbulence, one may describe the vortex in the source as a random variable. We simplify the vortex filament as the collection of particles with random locations. In this context, one may identify the particle's trajectory as the streamline of the fluid particle. We refer to Section 6.2 for more discussions.

As was discussed in the above, the real turbulent phenomena is a very complicated one, which involves many vortices (eddies) with different size and vorticity, or many sources and sinks with different quantity of fluid (charge) $Q$ coming in and out per unit time. Here, we consider a simple toy model, in which there exists a single dipole (with a single source and sink). More explicitly, the locations of sink and source is identified, having a constant dipole moment, $d_H$, but we keeps the essential ingredient of random modulation which can be stated in other word, the magnitude of the dipole moment $d_H = |\boldsymbol{d}_H|$ is fixed time-independently, while the direction of the moment, $\hat{\boldsymbol{d}}_H(t) = \boldsymbol{d}_H(t)/d_H$ is randomly (stochastically) modulated. That is, we focus on a special case where the locations and the charges of the source are,

$$\boldsymbol{\zeta}_1 = \boldsymbol{\zeta}, \quad \boldsymbol{\zeta}_2 = -\boldsymbol{\zeta}, \quad Q_1 = -Q_2 = -Q\,, \tag{6}$$

and take a limit

$$|2\boldsymbol{\zeta}(\boldsymbol{t})| \to 0, \quad Q \to \infty \quad \text{such that} \quad \boldsymbol{d}_H(t) = \frac{2Q}{2\pi}\boldsymbol{\zeta} = d_H \times \hat{\boldsymbol{d}}_H(t) \text{ finite.} \tag{7}$$

The equation (3) becomes,

$$\frac{d\boldsymbol{x}(t)}{dt} = \frac{d_H}{r^D}\left(\hat{\boldsymbol{d}}(t) - D\hat{\boldsymbol{x}}(t)\left(\hat{\boldsymbol{x}}(t) \cdot \hat{\boldsymbol{d}}(t)\right)\right). \tag{8}$$

The second term represents the velocity induced by a dipole. The vector $\hat{\boldsymbol{d}}(t) \in \mathbb{R}^D$ is a random variable with a fixed normalization $|\hat{\boldsymbol{d}}(t)| = 1$.

As we will show numerically in the next section, the particle's trajectory has a fractal dimension that is different from the normal Brownian motion. In the companion paper [32], we derive a Fokker-Planck equation associated with this stochastic differential equation. It analytically demonstrates the multi-fractal nature of the trajectory with different behaviors in the radial and angular directions. These observations seem to imply the relevance of this simple model to capture some nature of turbulence.

## 3 Simulation of trajectories: setup and summary

### Numerical simulation setup

We have used the 4-th order adaptive Runge-Kutta method in our numerical computations [33]. We took the error parameter $\epsilon = 0.0001$, smaller than the initial time step $dt = 0.01$. Roughly speaking, the time step $\Delta t$ is adjusted so that the error in the velocity field $V(x,t)$ can be the order of $\epsilon = 0.0001$, satisfying $|\Delta x/\Delta t| = |V(x,t)| = O(\epsilon)$. In this way, the time step becomes smaller, when the velocity field becomes larger near the dipole location.

The 4-th order Runge-Kutta formula gives the next position $x_{n+1}$, starting from position $x_n$ at time $t_n$, as follows:

$$z_i = x_n + \Delta t \sum_{j=1}^{i-1} a_{i,j} V(z_j, t_n + c_j \Delta t), \ i = 1, \ldots, s \tag{9}$$

$$x_{n+1} = x_n + \Delta t \sum_{j=1}^{s} b_j V(z_n, t_n + c_j \Delta t), \tag{10}$$

where $a_{i,j}$, $c_j$ and $b_j$ are chosen as are listed in the Butcher tableau, so that the truncation error satisfies $\mathcal{O}(\Delta t^{s+1})$. In this paper we choose $s = 4$.

| $c_j$ | $a_{i,j}$ | | | | | |
|---|---|---|---|---|---|---|
| 0 | | | | | | |
| $\frac{1}{4}$ | $\frac{1}{4}$ | | | | | |
| $\frac{3}{8}$ | $\frac{3}{32}$ | $\frac{9}{32}$ | | | | |
| $\frac{12}{13}$ | $\frac{1932}{2197}$ | $\frac{-7200}{2197}$ | $\frac{7296}{2197}$ | | | |
| 1 | $\frac{439}{216}$ | $-8$ | $\frac{3680}{513}$ | $\frac{-845}{4104}$ | | |
| $\frac{1}{2}$ | $\frac{-8}{27}$ | $2$ | $\frac{-3544}{2565}$ | $\frac{1859}{4104}$ | $\frac{-11}{40}$ | |
| | $\frac{16}{135}$ | $0$ | $\frac{6656}{12825}$ | $\frac{28561}{56430}$ | $\frac{-9}{50}$ | $\frac{2}{55}$ | $b_i$(5−th order's $b_i$) |
| | $\frac{25}{216}$ | $0$ | $\frac{1408}{2565}$ | $\frac{2197}{4104}$ | $\frac{-1}{5}$ | $0$ | $b_i$(4−th order's $b_i$) |

Table 1: The Runge-Kutta-Fehlberg method's Butcher tableau.(5-th and 4-th orders)

To estimate errors in $x_{n+1}$, we have to know the real solution after $t_n$, but we can not do it. Instead, what we can do is to take the solution $\hat{x}_{n+1}$ obtained in the fifth order Runge-Kutta method as a substitute for the real solution. This is called the Fehlberg method.

Thus, the local errors is estimated as

$$\mathrm{LE}_{n+1} \approx \hat{x}_{n+1} - x_{n+1}. \tag{11}$$

Then, the next time $t_{n+1}$ can be adjusted so that it may satisfy

$$0.5 \times \epsilon(t_{n+1} - t_n) < |\mathrm{LE}_{n+1}| < \epsilon(t_{n+1} - t_n). \tag{12}$$

Then new $\Delta t \to \Delta t'$ is

$$\Delta t' = 0.8 \Delta t \left( \frac{\epsilon}{|\mathrm{LE}_{n+1}|} \right)^{\frac{1}{5}}. \tag{13}$$

That is, when (12) is satisfied, the next $\Delta t'$ is determined by (13). If (12) is not satisfied, apply (13) until it is satisfied. This is the way to control errors in solving the ordinary differential equation by the adaptive Runge-Kutta method.

However, our paper deals with the stochastic differential equations with a random variable. In this case, the numerical estimation includes the statistical errors, depending on the number of trials $N$. Therefore, the error bar depicted in the tables and figures of the fractal dimensions $D_f$, consists of 1) the statistical error coming from the rolling of the dice $N$ times, 2) the error coming from the linear regression in estimating the fractal dimension by the box-counting method, in addition to 3) the error in the adaptive Runge-Kutta method. To estimate the errors, we set $N = 150$ for 1), for 2) we used a measure of the goodness of fit of the regression model called the coefficient of determination. It is worst at 0 and best

at 1. In this case, we set the model so that the coefficient of determination is greater than 0.8. And for 3) we set $\epsilon = 0.0001$.

The direction of the dipole moment, $\hat{\boldsymbol{d}}(t_i)$ $(i = 1, \cdots, N)$ is randomly modulated at each step $t_i$, while keeping $|\hat{\boldsymbol{d}}(t_i)| = 1$.

$\hat{\boldsymbol{d}}(t_i)$ is a homogeneous probability distribution and is generated by the following algorithm.

$$\hat{\boldsymbol{d}}(t_i) = \begin{pmatrix} a \\ b \\ c \end{pmatrix} \equiv \begin{pmatrix} \pm\sqrt{1 - \cos^2\theta}\cos\phi \\ \pm\sqrt{1 - \cos^2\theta}\sin\phi \\ \cos\theta \end{pmatrix} \tag{14}$$

where $\cos\theta$ is a uniform random variable from $-1$ to $1$, and $\phi$ is a uniform random variable from $0$ to $\pi$, where $\pm$ means there are two branches; $+$ branch gives $1/2$ probability of being positive, and $-$ branch does $1/2$ probability of being negative.

The discretized version of (8) becomes,

$$\boldsymbol{x}(t_i + dt) = \boldsymbol{x}(t_i) + \frac{d_H}{r^D(t_i)}dt\left(\hat{\boldsymbol{d}}(t_i) - D\hat{\boldsymbol{x}}(t_i)\left(\hat{\boldsymbol{x}}(t_i) \cdot \hat{\boldsymbol{d}}(t_i)\right)\right) \quad (i = 1, \cdots, N) \tag{15}$$

where $\hat{\boldsymbol{x}}(t_i) = \frac{\boldsymbol{x}(t_i)}{r(t_i)}$. We use the dipole moment $d_H$ as a parameter. The dimension $D$ is set to 2 or 3. The particle moves in a cubic region $\boldsymbol{x} \in [-L_f, L_f]^D$. The boundary condition will be important, and we will explain it in detail later.

## Fractal dimensions

We estimate the fractal dimension $D_f$ of the trajectory by the Box-Counting method: we divide the whole square (cubic) region by smaller boxes with the edge lengths $(\delta L) = (L, L/2, L/2^2, \cdots)$. For each division $\delta L$, we count the number $n$ of boxes that contain the trajectory. The fractal dimension $D_f$ is defined by

$$D_f = \lim_{\delta L \to 0} \frac{\ln n}{\ln \delta L}. \tag{16}$$

We estimate the ratio by the linear regression. When determining the fractal dimension, we used the coefficient of determination. In this case, we set the model so that the coefficient of determination is greater than 0.8.

## Boundary conditions

By a simple inspection of eq.(8), there is a solid bouncing effect in the vicinity of the dipole $(r = 0)$. It implies that the property of the system is sensitive to the choice of the boundary condition. Among various possible choices, we choose the following two boundary conditions. In both cases, we obtain an anomalous fractal dimension, which is different from that of a simple random walk.

### Condition 1: Periodic boundary condition

The first is the periodic boundary condition imposed on (the boundary of) the region $\boldsymbol{x}(t_i) \in [-L_f, L_f]^D$.

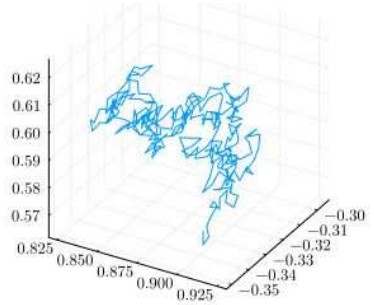

Figure 1: A trajectory of a particle when $L_f = 1.0$ imposing periodic boundary condition on 3D coordinates.

Figure 1 shows a particle trajectory with $L_f = 1.0$, when imposing the periodic boundary condition. In this case, the particle which jumps out of the fundamental region $\boldsymbol{x}(t_i) \in [-L_f, L_f]^D$ is brought back by the boundary condition. Such jumps over the lattices make the trajectory like Lévy Flight[12], which is a typical feature of the system with anomalous fractal dimension. Lévy Flight is discussed in detail in subsection 6.4.

## Condition 2: "Get back" to the original point

As the second choice, we impose the particle which bounces out of the fundamental region to return to the initial position of the particle. It was used by Mandelbrot to study fractal physics and sometimes referred to as resetting protocol[34]. We refer to such a boundary condition as Condition 2. We repeat it for a fixed time and determine the fractal dimension by the overlaid trajectories. This choice of the boundary condition will be helpful to study the relationship between the particle's initial position and the fractal dimension.

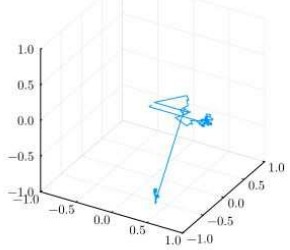

(a) Trajectory of a single particle with Condition 2, in which it starts from an intermediate position on 3D coordinates.

(b) Trajectories of particles with Condition 2, in which they start from same positions. These trajectories are colored on 3D coordinates.

Figure 2: Trajectories of a particle when $L_f = 1.0$ imposing Condition 2, the condition to restart from the initial position in case of going outside the boundary.

In this setup, we find some trajectories in Figure 2(b) where the particle is bounced away after moving to the vicinity of the dipole. On the other hand, the trajectory of a particle that starts from points far enough from the origin becomes a regular Gaussian random walk.

The particle starting from the intermediate position behaves to have both features (Figure 2(a)).

## Parameters

We use three parameters to perform the numerical simulation, the dipole moment $d_H$, the total number of steps $N$ and the box size $2L_f$. The range of each parameter is

$$5.0 \leq d_H \leq 100.0, \tag{17}$$
$$2.5 \times 10^4 \leq N \leq 5.0 \times 10^5, \tag{18}$$
$$0.025 \leq L_f \leq 0.5. \tag{19}$$

## Overview of numerical results

We will discuss the details in the next section. At first, we summarize the results obtained from the numerical calculations in Figure 3. In these Tables, $d_H$, $D_f$, and $\sigma$ imply the dipole moment, the mean value of the fractal dimension, and the standard deviation of the fractal dimension, respectively. Each table explains the behavior of the fractal dimension for different values of the dipole moment. The parameter regions to the results in Figure 3 are as follows: In Figure 3, $5.0 \leq d_H \leq 100.0$, $x_0 = -0.02, y_0 = 0.0, z_0 = 0.0, N = 1.0 \times 10^5$. $L_f = 0.1$ in Condition1. In Condition 2, $L_f = 1.0$

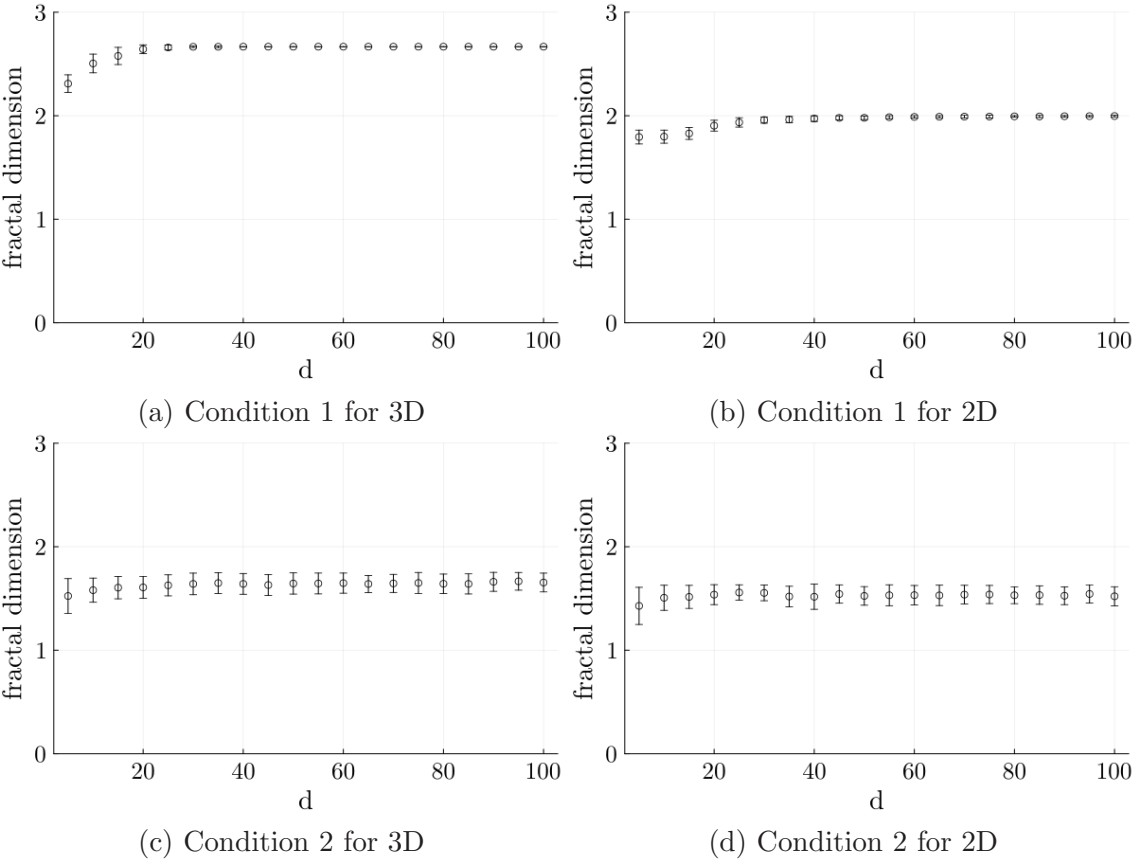

(a) Condition 1 for 3D     (b) Condition 1 for 2D

(c) Condition 2 for 3D     (d) Condition 2 for 2D

Figure 3: Dependence of $D_f$ on $d_H$ while keeping $N = 1.0 \times 10^5$, and $x_0 = -0.02, y_0 = 0.0, z_0 = 0.0$. In (a) and (b), $L_f = 0.1$. In (c) and (d), $L_f = 1.0$

Suppose the L'evy flight is realized by a big jump from the neighbor of the dipole to the neighbor of boundaries, we have approximately the [L'evy flight Condition (LfC)] as

$$\delta\boldsymbol{x}(t) = \frac{d_H}{(2L_f)^D}\delta t\,\boldsymbol{\Theta} \approx 2L_f, \ \ \text{or} \ \ [\text{LfC}] \equiv \frac{d_H\,\delta t}{(2L_f)^{(D+1)}} \approx 1. \tag{20}$$

where the average $\boldsymbol{\Theta} = \langle\hat{\boldsymbol{d}}(t) - D\hat{\boldsymbol{x}}(t)(\hat{\boldsymbol{x}}(t)\cdot\hat{\boldsymbol{d}}(t))\rangle$ is the order of 1. This affords a rough understanding of the parameter regions which give a stable fractal dimension $D_f$.

In Condition 1 (Periodic Boundary Condition), we estimate the fractal dimension of the trajectory at about 1.9 (2D) and about 2.7 (3D) for a wide range of the dipole moment. On the other hand, in Condition 2 (Get Back), the fractal dimension is estimated to be about $D_f \sim 1.6$ for 3D and $D_f \sim 1.5$ for 2D. Namely, we obtain slightly smaller fractal dimensions for Condition 2. As detailed in the next section, the fractal dimension behaves in a similar pattern in 2D and 3D cases, and the degree of decrease for $D_f$ depends only on $d_H$.

# 4   Details of the numerical simulation

In this section, we summarize the dependence of the fractal dimension $D_f$ on $N$ (the number of steps) and $L_f$ (the box size). We also analyze the number of particles which are bumped away from the fundamental domain if we do not impose the boundary conditions. Finally, we study the dependence on the initial location of the particle.

# Dependence on $N$, the number of steps

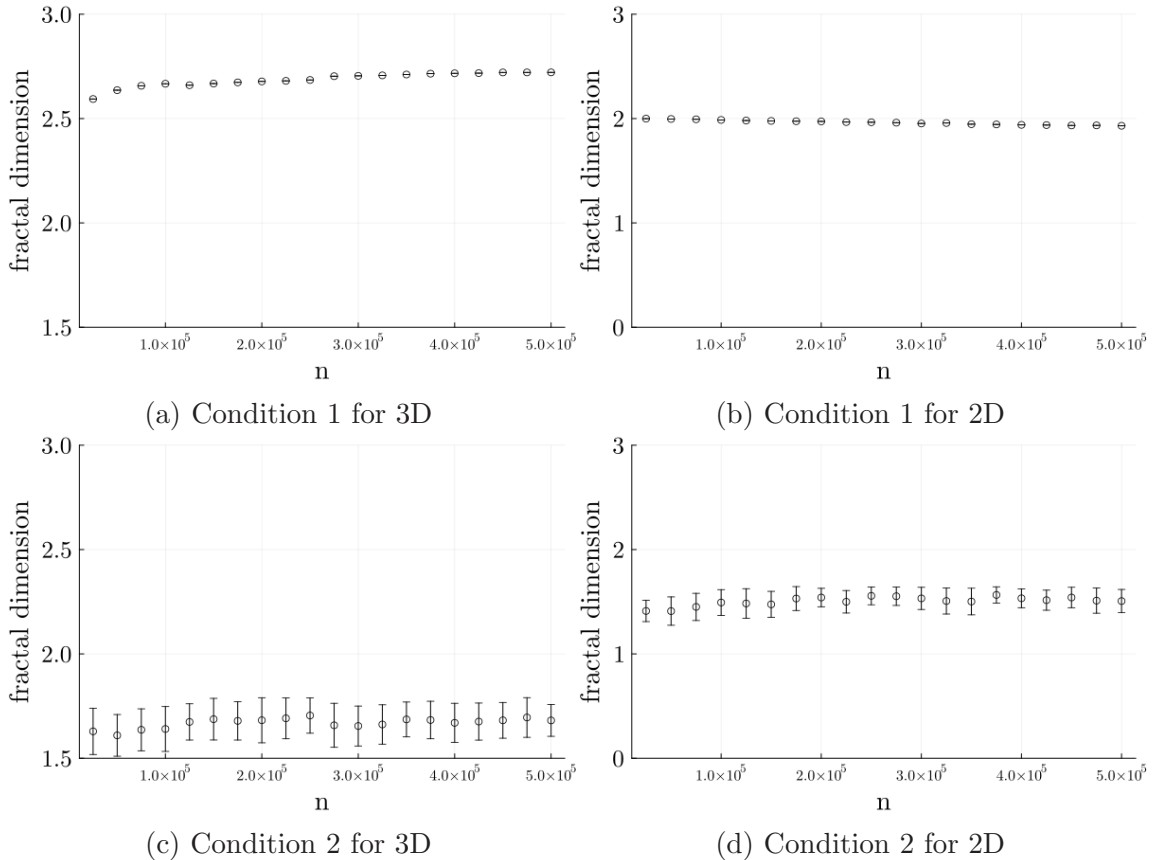

(a) Condition 1 for 3D        (b) Condition 1 for 2D

(c) Condition 2 for 3D        (d) Condition 2 for 2D

Figure 4: Dependence of $D_f$ on $N$ while keeping $d_H = 60.0$, and $x_0 = -0.02, y_0 = 0.0, z_0 = 0.0$. In (a), (b), and (d), $L_f = 0.1$. In (c), $L_f = 1.0$

When the parameter $N$ (the number of steps) varies as $2.5 \times 10^4 \le N \le 5.0 \times 10^5$, while keeping $d_H = 0.1$, $L_f = 0.1$(Condition 1 for 3D and 2D,Condition 2 for 2D) or $L_f = 1.0$(Condition 2 for 3D), and $x_0 = -0.02, y_0 = 0.0, z_0 = 0.0$ the fractal dimension $D_f$ asymptotically approaches a constant, which is equal to the values given in the previous section. Even if the scale invariance is not broken strongly by the power of $N$, it is weakly broken in detail. We accept these corrections and understand that the fractal dimension $D_f$ is approximately constant. And we use the box-counting method here, but the need to increase the number of divisions as $N$ increases is difficult from the standpoint of computation time when $N$ is large enough. If the number of divisions cannot be increased as $N$ increases, the fractal dimension of the particle trajectory will asymptotically approach the spatial dimension since it covers the box. This is a technical issue, as values of $N$ that are too large do not give reliable results, and for such $N$, calculations need to be made with a more detailed number of divisions.

# Dependence on the box size $L_f$

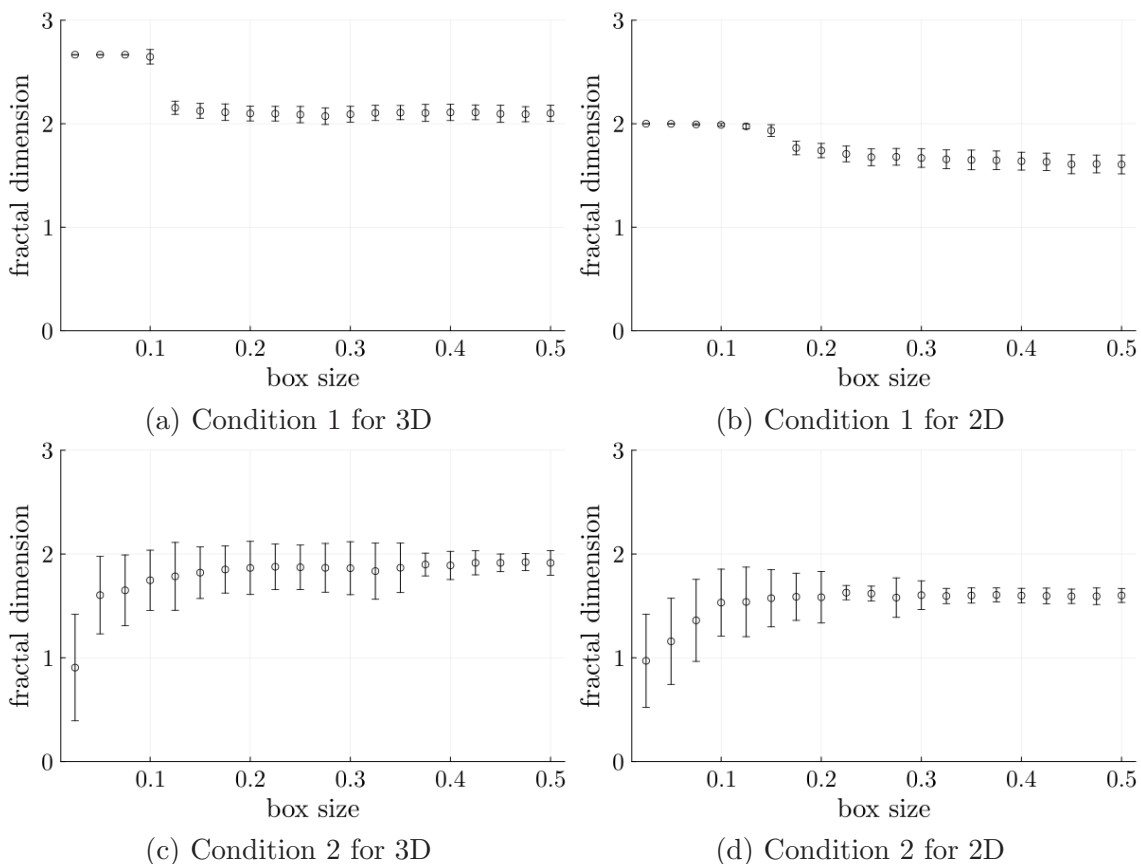

(a) Condition 1 for 3D

(b) Condition 1 for 2D

(c) Condition 2 for 3D

(d) Condition 2 for 2D

Figure 5: Dependence of fractal dimension $D_f$ on $L_f$ while keeping $d_H = 60.0$, $N = 1.0 \times 10^5$, and $x_0 = -0.02, y_0 = 0.0, z_0 = 0.0$.

When the box length $L_f$ varies as $0.025 \leq L_f \leq 0.5$, while keeping $d_H = 60.0$, $N = 1.0 \times 10^5$, and $x_0 = -0.02, y_0 = 0.0, z_0 = 0.0$ the fractal dimension has a plateau at the values 2.7 (Condition 1 for 3D), 1.9 (Condition 1 for 2D) which are consistent with the summary in Section 5.

On the other hand, in Condition 2, the fractal dimension asymptotically approaches 1.6 – 1.9 (Condition 2 for 3D), or 1.5 (Condition 2 for 2D).

## Missing particles

So far, we have used the boundary conditions by which we recover the particles bounced off the fundamental domain. In the analytical study performed in the companion paper [32], we will not recover these particles. For comparison, we study how particles decrease when we do not recover the bounced particles, depending on the step number $N$.

Figure 6 gives the step (time) dependence of the number of particles in 3D and 2D for $d_H = 1.0$ and $L_f = 3.0$ and the initial position $x_0 = 1.0$. We observe that particles number decreases exponentially in both cases. Therefore, for the value $N = 1.0 \times 10^5$ where we have observed the fractal dimension, most of the particles are restored ones through the boundary conditions. Therefore, the choice of Condition 1 or 2 is irrelevant to this calculation since we are only counting the number of particles out of the box.

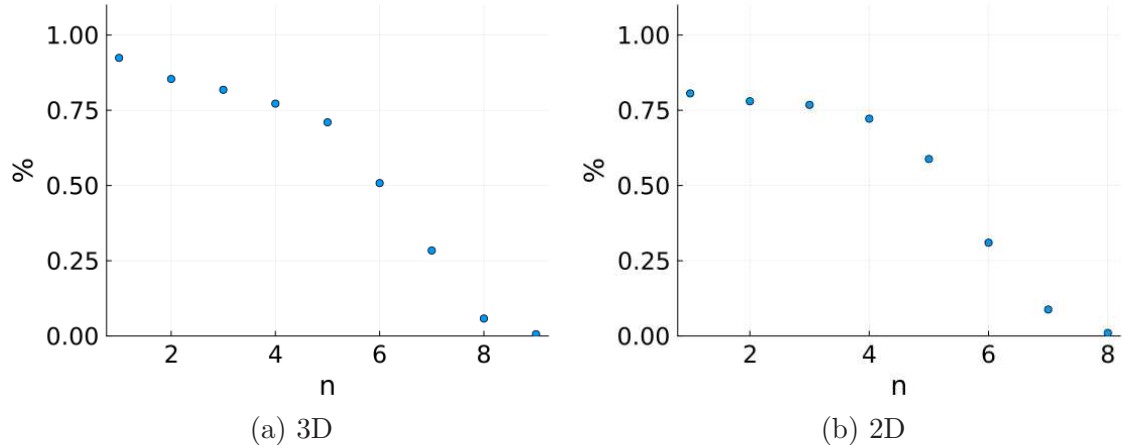

(a) 3D    (b) 2D

Figure 6: Decrease of particles while keeping $d_H = 1.0$ and $L_f = 3.0$ and the initial position $x_0 = 1.0, y_0 = 0.0, z_0 = 0$ and Number of steps $= 2^n$

## Dependence on the initial location $x_0$

Finally, we explain the dependence of the fractal dimension on the initial location $x_0$. One may naively expect that the fractional dimension is not well-defined when the initial particle location is too close to the dipole. In Figure 7, we show the plot of the fractal dimension in $D = 3$ with $N = 10^5$, various choices of $d_H$ and $L_f$, and use Condition 2.

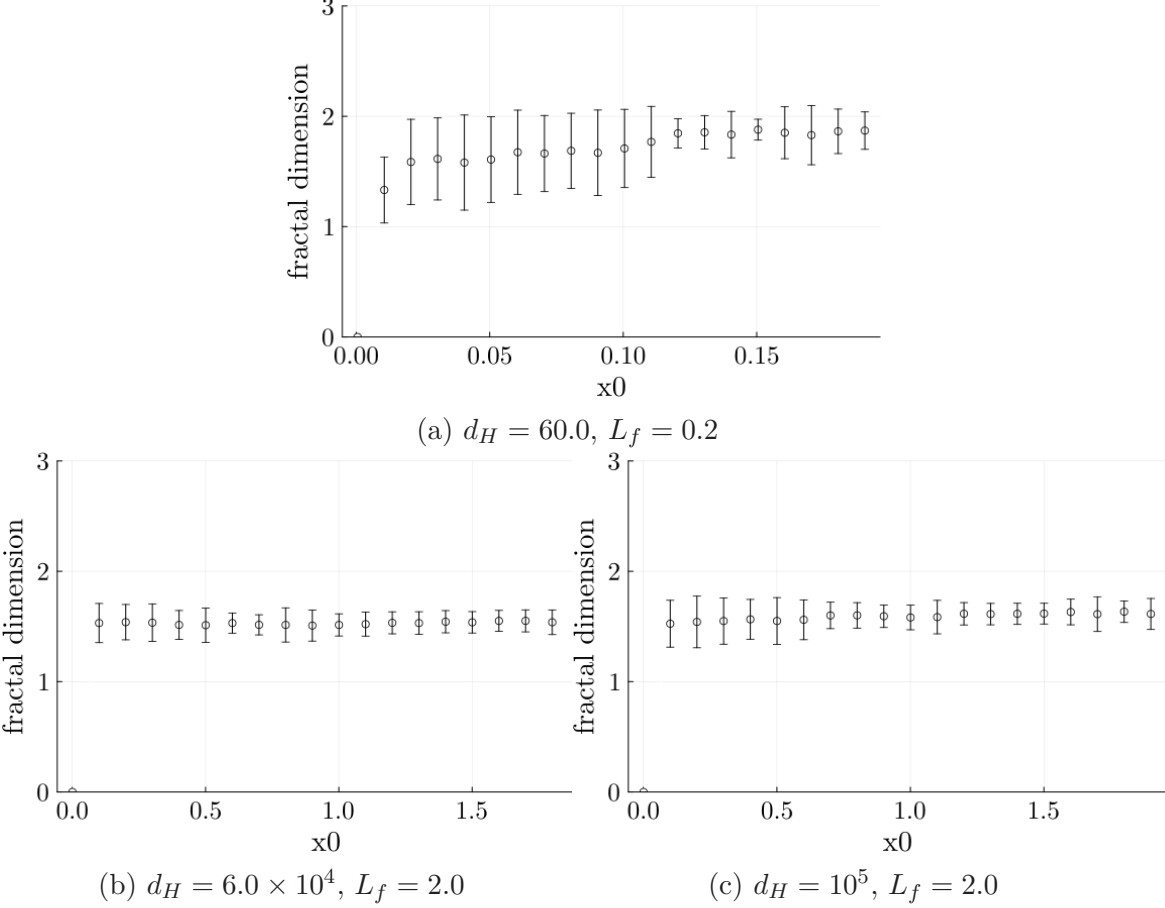

(a) $d_H = 60.0$, $L_f = 0.2$

(b) $d_H = 6.0 \times 10^4$, $L_f = 2.0$

(c) $d_H = 10^5$, $L_f = 2.0$

Figure 7: Dependence on fractal dimension $D_f$ on initial location $x_0$ while keeping $N = 10^5$, $y_0 = 0, z_0 = 0$, and Condition 2

## Replacing from $r$ to $r + \Delta r$

In order to perform calculations around the singularity, it is possible to include the cut-off in the denominator. Introduce the cut-off to eq. (15),

$$\boldsymbol{x}(t_i + dt) = \boldsymbol{x}(t_i) + \frac{d_H}{(r(t_i) + \Delta r)^D} dt \left( \hat{\boldsymbol{d}}(t_i) - D\hat{\boldsymbol{x}}(t_i)\left( \hat{\boldsymbol{x}}(t_i) \cdot \hat{\boldsymbol{d}}(t_i) \right) \right) \quad (i = 1, \cdots, N), \quad (21)$$

where $\Delta r$ is the cut-off.

We also examined the cut-off contribution when varying boxsize $L_f$ in the Condition 1 and Condition 2 cases, and in the 2D and 3D cases, respectively.

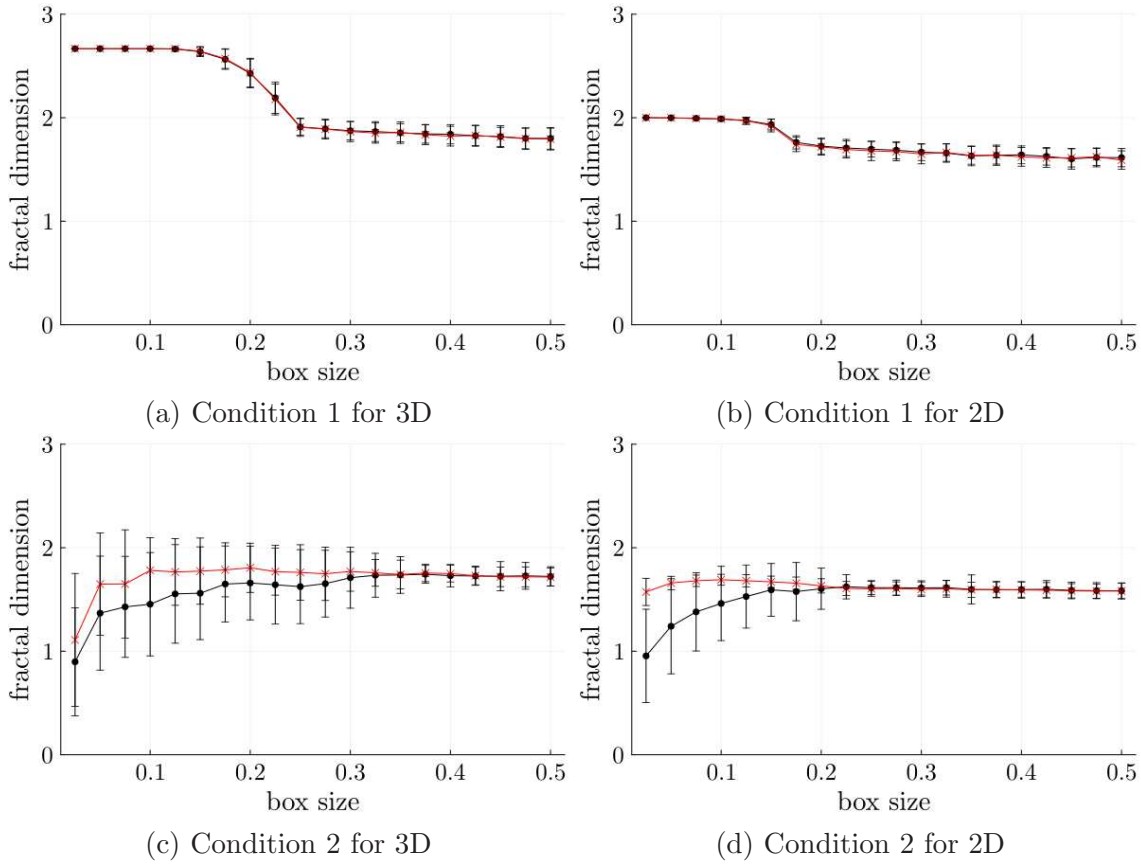

(a) Condition 1 for 3D

(b) Condition 1 for 2D

(c) Condition 2 for 3D

(d) Condition 2 for 2D

Figure 8: Dependence of fractal dimension $D_f$ on $L_f$ and cut-off $\Delta r = 0.0, 0.001$ while keeping $d_H = 60.0$, $N = 1.0 \times 10^5$, and $x_0 = -0.02, y_0 = 0.0, z_0 = 0.0$.

Figure 8 shows black line is $\Delta r = 0.0$ and red line is $\Delta r = 0.001$ while keeping $d_H = 60.0$, $N = 1.0 \times 10^5$, and $x_0 = -0.02, y_0 = 0.0, z_0 = 0.0$.

The discrepancy appearing in the fractal dimension $D_f$ in Condition 2, for different choices of cut-off $\Delta r$, is within the error bar $(1\sigma)$, so that it is not physically important at this level.

# 5    Summary

We performed a simulation about a particle motion in the potential of a dipole located at the origin in two and three dimensions. The direction of the moment of dipole varies randomly at each step, which gives rise to a stochastic differential (difference) equation of motion. We have estimated the fractal dimension $D_f$ of the trajectories in 2D and 3D models with the dipole moment $d_H$, the total number of steps $N$, and the box size $2L_f$ as the parameters. The trajectory behaves in a similar pattern to Lévy Flight by bouncing off the dipole and the boundary conditions of the system. The simulation of the two boundary conditions gives the fractal dimension of the trajectory be 1.5–1.9 (2D) and 1.6–2.7 (3D)[1] for a wide range of the parameters. These results are obtained in Condition 1 (Periodic

---

[1]For 2D and 3D Condition 1, we find the behaviour of a particle depending on the parameter $\alpha(d_H, \Delta t, L_f) = \frac{d_H \Delta t}{L_f^D}$. At this time, we get the maximum value of the fractal dimension $D_f \sim 1.9$ (2D) and $D_f \sim 2.7$ (3D) when $N = 1.0 \times 10^5$ is fixed.

boundary condition). The fractal dimension estimated in Condition 2 (Get back to the original points), gives 1.6 – 1.9 (3D), or 1.5 (2D). In the companion paper [32], we give an analytical estimation of the fractal dimension by obtaining and solving the Fokker-Planck equation associated with the model.

This dipole model adds a physical picture of anomalous diffusion, which we hope will be helpful in turbulence research. For this purpose, a discussion related to fluid dynamics is given in the following discussion section.

# 6    Discussions

We will give some theoretical prospects of various topics, which motivate the paper. While they are not within reach of this paper, they will be important in future research.

## 6.1    Modulation by vortex

In this paper, we examined a toy model of hydrodynamics, where a randomly modulated dipole at the origin determines the velocity field in a finite box with edge length $2L_f$. It models a picture that the emergence of eddies (or vortices) is responsible for the generation of turbulence. We simplified the eddies' modulation to that of a dipole in this paper. To investigate a more realistic case of eddies' modualtion, we will refer to eqs.(15) and (17) in [23]. Then, we can discuss as follows, by restricting to a point-like vortex in 2D and a vortex filament in 3D. In 2D, the circulation $\Gamma$ of a point-like vortex located at $\boldsymbol{X}(t)$ reads

$$\omega(t, \boldsymbol{x}) = (\boldsymbol{\nabla} \times \boldsymbol{v})_3 = \partial_1 v_2 - \partial_2 v_1 = \Gamma \, \delta^{(2)}(\boldsymbol{x} - \boldsymbol{X}(t)). \tag{22}$$

In 3D, we have to introduce $\boldsymbol{X}(t, \sigma)$ to express the location of a vortex filament; the extension of the filament is parametrized by $\sigma$. Then, the circulation is given by

$$\boldsymbol{\omega}(t, \boldsymbol{x}) = \boldsymbol{\nabla} \times \boldsymbol{v} = \Gamma \int d\sigma \, \frac{\partial \boldsymbol{X}}{\partial \sigma} \delta^{(3)}(\boldsymbol{x} - \boldsymbol{X}(t, \sigma)). \tag{23}$$

Now, the motion of a fluid particle can be studied in the background field of the random modulation of the vortices. This motion corresponds to the motion of a charged particle in the background field of the random distribution of magnetic strings. The velocity field is obtained as

$$\begin{cases} \boldsymbol{V}_\zeta(\boldsymbol{x}; t) = Q_0 \sum_i \frac{\Gamma_i}{2\pi} \frac{(-(x_2 - X_2(t,i))(x_1 - X_1(t,i)))}{|\boldsymbol{x} - \boldsymbol{X}(t,i)|^2}, \text{ for 2D}, \\ \boldsymbol{V}_\zeta(\boldsymbol{x}; t) = Q_0 \sum_i \frac{\Gamma_i}{4\pi} \int d\sigma_{,i} \frac{\partial \boldsymbol{X}(t,\sigma_{,i})}{\partial \sigma_{,i}} \times \frac{\boldsymbol{x} - \boldsymbol{X}(t,\sigma_{,i})}{|\boldsymbol{x} - \boldsymbol{X}(t,\sigma_{,i})|^3}, \text{ for 3D}. \end{cases} \tag{24}$$

The issue is how to introduce the proper configuration of vortices (eddies) and modulate the vortices randomly. We have to refer to [35, 36, 37, 38, 39, 40, 41, 42, 43], but this issue is not an easy one, since it involves lots of unsolved problems.

## 6.2    Navier-Stokes equation and averaged effect of random modulation of dipoles on it

Corresponding to the streamlines, $\boldsymbol{x} = \overline{\boldsymbol{x}}(t)$, for which we performed the simulation in Section 3, we have the following Navier-Stokes (N-S) equation:

$$\text{(Navier-Stokes)} : \partial_t \boldsymbol{V}_\zeta(\boldsymbol{x}; t) + (\boldsymbol{V}_\zeta(\boldsymbol{x}; t) \cdot \boldsymbol{\nabla}) \, \boldsymbol{V}_\zeta(\boldsymbol{x}; t) = -\frac{1}{\rho} \boldsymbol{\nabla} p + \nu \Delta \boldsymbol{V}_\zeta(\boldsymbol{x}; t). \tag{25}$$

With this equation, we can estimate the averaged effect of the random modulation of $N_{\text{dipole}}$ dipoles on it. Taking the spatial dimensions $D$ to be arbitrary, the vector indices $(\mu, \nu)$ run from 1 to $D$. Suppose $\boldsymbol{v}(\boldsymbol{x}, t)$ be a generic velocity field with no random modulation of dipoles. The velocity field, after the dipole modulation is applied to the system, can be

$$\boldsymbol{V}_{\hat{\zeta}(t)}(\boldsymbol{x}, t)^{\mu} = \boldsymbol{v}(\boldsymbol{x}, t)^{\mu} + d_H \sum_{i=1}^{N_{\text{dipole}}} \sum_{\nu=1}^{D} \frac{1}{r_i^D} T_i^{\mu\nu} \hat{\boldsymbol{d}}_i(t)^{\nu}, \quad \text{with} \quad \hat{\boldsymbol{\zeta}}_i(t) = \hat{\boldsymbol{d}}_i(t), \quad (26)$$

where $T^{\mu\nu}$ is

$$T_i^{\mu\nu} = \delta^{\mu\nu} - D \, \hat{\boldsymbol{r}}_i^{\mu} \hat{\boldsymbol{r}}_i^{\nu}, \quad (27)$$

a symmetric and traceless tensor, $T^{\mu\nu} = T^{\nu\mu}$, $\text{tr}T = 0$, having the anguler momentum 2.

Following [44], we have arrived at the N-S equation modified by the random modulation of dipoles:

$$\partial_t \boldsymbol{v}(\boldsymbol{x}; t) + (\boldsymbol{v}(\boldsymbol{x}; t) \cdot \boldsymbol{\nabla}) \, \boldsymbol{v}(\boldsymbol{x}; t) = -\boldsymbol{\nabla}p + \nu \Delta \boldsymbol{v}(\boldsymbol{x}; t) + d_H^2(D-1) \sum_{i=1}^{N_{\text{dipole}}} \frac{\boldsymbol{x} - \boldsymbol{x}_i}{|\boldsymbol{x} - \boldsymbol{x}_i|^2}. \quad (28)$$

To make clear the meaning of the last term proportional to $d_H^2$, we have to solve the modified N-S equation explicitly. The following is the derivation of Eq.(28): The equation Eq. (26) can be considered as the "constitutive equation" in non-equilibrium thermodynamics. Then, we can apply the standard method of non-equilibrium thermodynamics developed by Onsager and Machlup and by Hashitsume for this averaging process (see [44]).

Now, we will examine the averaging of the Navier-Stokes equation over the random modulation of dipoles. If the averaging over the direction of dipole $\hat{\boldsymbol{\zeta}} = \hat{\boldsymbol{d}}$ is performed, which is denoted by $\langle ... \rangle_{\hat{\zeta}}$, we have

$$\langle \partial_t \boldsymbol{V}_{\zeta}(\boldsymbol{x}; t) + (\boldsymbol{V}_{\zeta}(\boldsymbol{x}; t) \cdot \boldsymbol{\nabla}) \, \boldsymbol{V}_{\zeta}(\boldsymbol{x}; t) \rangle_{\hat{\zeta}} = -\boldsymbol{\nabla}p + \langle \nu \Delta \boldsymbol{V}_{\zeta}(\boldsymbol{x}; t) \rangle_{\hat{\zeta}}, \quad (29)$$

where the averaging is expressed by

$$\langle O(\boldsymbol{x}, \zeta(t)) \rangle_{\hat{\zeta}} = \frac{1}{(A_D)^{N_{\text{dipole}}}} \int \prod_{i=1}^{N_{\text{dipole}}} \mathcal{D}\hat{\boldsymbol{\zeta}}_i(t) \, O(\boldsymbol{x}, \zeta(t)), \quad (30)$$

$A_D$ is the surface area of a unit ball in $D$ spatial dimensions; $A_2 = 2\pi$, $A_3 = 4\pi$, $\cdots$, and

$$\langle \hat{\zeta}_i^{\mu} \rangle_{\hat{\zeta}} = 0, \quad \langle \hat{\zeta}_i^{\mu} \hat{\zeta}_j^{\nu} \rangle_{\hat{\zeta}} = \frac{1}{D} \delta_{ij} \delta^{\mu\nu}. \quad (31)$$

In the averaging, $\boldsymbol{V}_{\zeta}(\boldsymbol{x}; t)$ can be replaced by $\boldsymbol{d}_i$ or $\boldsymbol{\zeta}_i$, using the constitutive equation. Thus, we have

$$\langle \boldsymbol{V}_{\zeta} \rangle_{\hat{\zeta}} = 0, \quad (32)$$

$$\langle \boldsymbol{V}_{\zeta}^{\mu} \boldsymbol{V}_{\zeta}^{\nu} \rangle_{\hat{\zeta}} = \boldsymbol{v}^{\mu} \boldsymbol{v}^{\nu} + \frac{d_H^2}{D} \sum_{i=1}^{N_{\text{dipole}}} \frac{\delta^{\mu\nu} + D(D-2)\hat{r}_i^{\mu}\hat{r}_i^{\nu}}{(r_i)^{2D}}. \quad (33)$$

The necessary formula here is

$$\langle (\boldsymbol{V}_\zeta \cdot \boldsymbol{\nabla}) \boldsymbol{V}_\zeta^\mu \rangle_{\hat\zeta} = (\boldsymbol{v} \cdot \boldsymbol{\nabla}) \boldsymbol{v}^\mu - d_H^2 (D-1) \sum_{i=1}^{N_{\rm dipole}} \frac{\hat{r}_i^\mu}{r_i}. \tag{34}$$

In this way, we arrive at the N-S equation including a correction from the random modulation of dipoles:

$$\partial_t \boldsymbol{v}(\boldsymbol{x};t) + (\boldsymbol{v}(\boldsymbol{x};t) \cdot \boldsymbol{\nabla}) \, \boldsymbol{v}(\boldsymbol{x};t) = -\boldsymbol{\nabla} p + \nu \Delta \boldsymbol{v}(\boldsymbol{x};t) + d_H^2(D-1) \sum_{i=1}^{N_{\rm dipole}} \frac{\boldsymbol{x} - \boldsymbol{x}_i}{|\boldsymbol{x} - \boldsymbol{x}_i|^2}. \tag{35}$$

The last term proportional to $d_H^2$ is the correction.

## 6.3  Energy dissipation rate per unit mass

The fractal dimensions which we have discussed in this paper, are deeply related to the scaling laws between $\boldsymbol{x}$ and $t$. Therefore, it is reasonable to examine our model in the light of the scaling laws by Kolmogorov and Kraichnan [22, 23, 24, 25, 26, 45, 46]. Here, we examine the energy dissipation rate per unit mass $\epsilon_r$.[2]

The $\epsilon_r$ can be estimated by relating it to [kinetic viscosity] × [squared shear strain tensor] [22, 23]. To make clearer the dependence of the rate on the distance $r$ from the position of a dipole, we have introduced $\epsilon_r$. This variable can represent the scaling property for $r/L_f$, depending on how we approach the singularity, the location of a dipole, where the continuity condition $\boldsymbol{\nabla} \cdot \boldsymbol{v} = 0$ is violated.

The energy dissipation rate per unit mass $\epsilon_r$ in 2D and 3D is given as follows:

$$\epsilon_r = \begin{cases} 8\pi \frac{\nu(d_H)^2}{L_f^3} r^{-4} & (2D), \\ \frac{288\pi}{5} \frac{\nu(d_H)^2}{L_f^2} r^{-5} & (3D). \end{cases} \tag{36}$$

This is a scaling law of our model. The energy dissipation rate per unit mass is not a constant as in Kolmogorov [22, 23] but depends on $r^{-4}$ in 2D and $r^{-5}$ in 3D, respectively.

More specifically, we will estimate $\epsilon_r$ at a length scale $r$, under the influence of the random modulation of a single dipole. The scale $r$ can be the distance from the dipole.

The velocity field $\boldsymbol{V}(\boldsymbol{x},t)$ is given under the dipole modulation as follows:

$$\boldsymbol{V}(\boldsymbol{x},t) = \boldsymbol{v}_0(\boldsymbol{x};t) + \frac{d_H}{r^D} \left\{ \hat{\boldsymbol{d}}(t) - D\hat{\boldsymbol{r}}(\hat{\boldsymbol{r}} \cdot \hat{\boldsymbol{d}}(t)) \right\} \quad (\text{in } D \text{ dimensions}). \tag{37}$$

The quantity to estimate is

$$\epsilon_r = \frac{\nu/2}{L_f^D} \int_r^\infty d^D\boldsymbol{r} \sum_{\alpha,\beta} (\partial_\alpha V_\beta + \partial_\beta V_\alpha)^2. \tag{38}$$

A straightforward calculation shows

$$\epsilon_r = \frac{2\nu(d_H)^2}{L_f^D} \int_r^\infty d^D\boldsymbol{r} \, \frac{1}{r^{2(D+1)}} \left( 2 + (D+1)(D-2)(\hat{\boldsymbol{r}} \cdot \hat{\boldsymbol{d}}(t))^2 \right). \tag{39}$$

The time averaging gives $\langle (\hat{\boldsymbol{r}} \cdot \hat{\boldsymbol{d}}(t))^2 \rangle_{\hat\zeta(t)} = \frac{1}{2}$, so that we arrive at the energy dissipation rate in Eq.(36).

---

[2]Here, we examine the energy dissipation rate per unit mass $\epsilon_r$

## 6.4   Relationship between fractal dimension and Lévy Flight

If the distribution function $p(x)$ has the non-vanishing but finite, second order moment[3], then the central limit theorem states that the asymptotic form of the distribution function is Gaussian. The power distribution, however, is given by

$$p(x) \propto \frac{A}{|x|^\alpha} + \dots, \tag{40}$$

which has the divergent second-order moments for $0 < \alpha < 2$.

Its characteristic function is

$$\lambda(k) = \int_{-\infty}^\infty dx \, p(x) e^{ikx} \tag{41}$$

$$= 1 - \tilde{A}|k|^\alpha + \dots, \tag{42}$$

$$\tilde{A} \equiv A \int_{-\infty}^\infty dy \frac{1 - \cos y}{|y|^{1+\alpha}}. \tag{43}$$

This is the distribution function of the Lévy flight. The probability to make $n$ times flights is

$$P_n(k) = \lambda(k)^n \approx e^{-n\tilde{A}|k|^\alpha}. \tag{44}$$

We counted the variations at each time step in our numerical simulation of the dipole model. We compare the obtained results with a Lévy flight model with infinite time steps, for $0 < \alpha' < 2$ which is described by

$$p(x) \propto \frac{e^{\sigma/2(x-\mu)}}{(x - \mu)^{1+\alpha'}}. \tag{45}$$

If $x$ is large $(|x - \mu| \gg \sigma)$ , we have the following expansion,

$$p(x) \propto \frac{1}{(x-\mu)^{1+\alpha'}} + O\left(\frac{1}{(x-\mu)^{2+\alpha'}}\right), \tag{46}$$

which gives the distribution function with power behavior.

The graphical comparison of the model with 3 dim numerical results for $D_f = 2.73$ is shown in Figure 5.1. This shows that the numerical results are consistent with the Lévy-Flight model with $\alpha = 2.0, \mu = 0.14$, and $\sigma = 0.9$.

The relation between the power law exponent $\alpha$ and the fractal dimension $D_f$ of the trajectory is used to define the fractal dimension in the box-counting method.

To understand the relation, one particle is assumed to be added at each time step $\Delta t = 1$. Then the total number of particles added on a trajectory is $t$. If these particles are distributed $D_f$ dimensional cubic box with the side length $\langle(\Delta x)^2\rangle^{1/2}$, then we have $(\langle(\Delta x)^2\rangle^{1/2})^{D_f} \propto t$, or equivalently $\langle(\Delta x)^2\rangle^{1/2} = \langle|\Delta x|\rangle \propto t^{1/D_f} \propto t^\alpha$. This is the restatement of the definition of the fractal dimension in the box-counting method, which gives $D_f = 1/\alpha$.

One more important relationship exists between $D_f$ and the scaling behavior of the correlation functions. For example, the 2-point correlation function is the Green function

---

[3] $\langle(x - \langle x\rangle)^2\rangle$, with $\langle O(x)\rangle \propto \int dx \, O(x) \, p(x)$

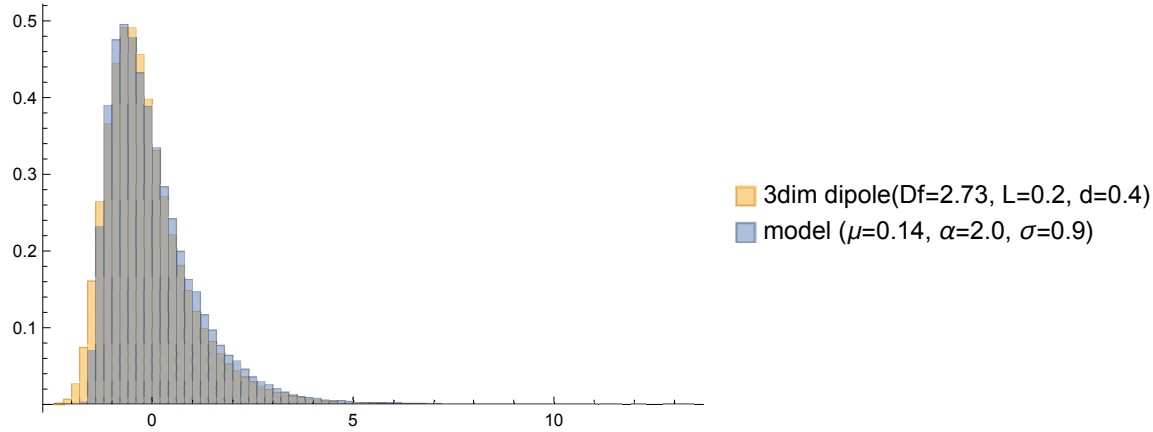

Figure 9: A graphical comparison of the model and 3 dim numerical results for $D_f = 2.73$ where the horizontal axis represents $x$ and the vertical axis represents $p(x)$

$G(x, 0)$, defined by $\hat{\mathcal{D}}G(x, 0) = \delta^{(D)}(x)$, where $\hat{\mathcal{D}}$ is the differential operator which appears in the diffusion equation, $\partial_t P(x; t) = K\hat{\mathcal{D}}P(x; t)$. As was shown by Einstein, this diffusion equation determines the probability distribution $P(x; t)$ of a random walk particle, starting from the origin and arriving at $x$ after $t$ time steps. The operator $\hat{\mathcal{D}}$ is not necessarily the Laplacian, but it can take more complicated differential operators.

If the 2-point function gives the power law, $\propto |x|^{-D+\beta}$, then, we know $t \propto |x|^{\beta}$, which means $D_f = \beta$. More generally, the scaling behavior (power behavior in the coordinate) of the $q$-point correlation functions gives scale dimension $|x|^{\tau(q)}$. This $\tau(q)$ may differ from $q(-D + D_f)$ as was known in the multi-fractals.

Therefore, the estimation of fractal dimension for the trajectory studied in this paper, and that for the various correlation functions, is important in various anomalous diffusion models such as hydrodynamics and Lévy walk.

# Acknowledgments

We would like to thank Prof. Fukumoto for the invitation to the workshop,"Helicity and space-time symmetry - a new perspective of classical and quantum systems", October 5-8 (2021), OCAMI, Osaka City University, where a part of the result was announced. We are indebted to Shiro Komata for reading this paper and giving useful comments. YM is partially supported by JSPS Grantin-Aid KAKENHI (#18K03610, JP21H05190).

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
