# Peer review of "Anomalous diffusion in a randomly modulated velocity field"

_SciPost Physics_

## Round 2 · Referee Report · Anonymous (Referee 1) · 2022-2-14

Strengths

The original approach to the long-standing, hardly resolvable turbulence problem

Weaknesses

The model is not clearly justified
Some typos

Report

  1. The authors investigated numerically anomalous diffusion in a randomly modulated dipolar velocity field to construct a toy model for an anomalous diffusion of fluid particles in turbulence. The origin of the model is explained in the discussion section 6. The authors calculated fractal dimensions of the trajectory in two and three dimensions for two different types of boundary conditions including periodic ones and “Get back” to the original point conditions.

The anomalous diffusion is, indeed, found. The fractal dimension is sensitive to the boundary conditions (needed because of ``a solid bouncing effect") and dipole moment. This work is an interesting approach to a long-standing problem of turbulence and it probably deserves publication since it can present a breakthrough. I have certain difficulties related mostly to the relevance of the model which, I wish to be addressed before the publication as described below.

  1. As I understand boundary conditions are needed because of the singularity in the dipolar field. Is this singularity, indeed, significant in liquids? My first try for the problem like that would be to cut off the field divergence at certain radius r=r_0 for instance, replacing r with r+r_0 in the denominator in Eq. 7. Then singularity will disappear. In that case the boundary conditions are not needed, and I would expect the fractal dimension to be equal to the space dimension. Can the authors comment on that and include this case into the consideration?

  2. I guess the word "of" is missed in the third line of the next to the last paragraph in Sec. I.

  3. In the last part of eq. 5 should it be vector in the right hand side.

4,. Table 3 contains zeros for dimensions in its second column. Why? Sorry if I missed explanations.

  1. In addition to the velocity field there can exist the ordinary brownian motion contribution to the liquid particle dynamics. Is this right to ignore it?

  2. Is there some way to verify the theory experimentally? Can the authors comment on that?

Requested changes

  1. As I understand boundary conditions are needed because of the singularity in the dipolar field. Is this singularity, indeed, significant in liquids? My first try for the problem like that would be to cut off the field divergence at certain radius r=r_0 for instance, replacing r with r+r_0 in the denominator in Eq. 7. Then singularity will disappear. In that case the boundary conditions are not needed, and I would expect the fractal dimension to be equal to the space dimension. Can the authors comment on that and include this case into the consideration?

  2. I guess the word "of" is missed in the third line of the next to the last paragraph in Sec. I.

  3. In the last part of eq. 5 should it be vector in the right hand side.

4,. Table 3 contains zeros for dimensions in its second column. Why? Sorry if I missed explanations.

  1. In addition to the velocity field there can exist the ordinary brownian motion contribution to the liquid particle dynamics. Is this right to ignore it?

  2. Is there some way to verify the theory experimentally? Can the authors comment on that?

  • validity: high
  • significance: high
  • originality: high
  • clarity: good
  • formatting: excellent
  • grammar: excellent

Author:  Yoshiki Matsuoka  on 2022-05-03  [id 2433]

(in reply to Report 1 on 2022-02-14)

We have resubmitted the paper and have attached our reply in the attached pdf.
Thank you for your cooperation.

Attachment:

reply_t1by0vr.pdf

Anonymous on 2022-05-18  [id 2496]

(in reply to Yoshiki Matsuoka on 2022-05-03 [id 2433])

I found the responses by the authors quite acceptable and in my opinion the necessary improvement is made. In my opinion the manuscript is ready for publication.

---

## Round 2 · Referee Report · Anonymous (Referee 2) · 2022-3-9

Strengths

1 - Interesting realization of a Levy flight.
2 - Possible relevance to turbulence.

Weaknesses

1 - Doubtful choice of boundary conditions.
2 - The definition of the fractal dimension for a finite-time particle trajectory in a finite-size system is not universal and highly system specific and parameter-dependent.
3 - Unclear or doubtful analysis of the results with respect to the parameters (system size, number of time steps, initial particle location)
4 - Non-standalone manuscript: many references to the results of a companion paper [23] and the other one of the authors [24].

Report

In this numerical paper the authors consider a model of a classical particle which random movement in a finite D-dimensional cube of a size $2L_f$ in each direction is mediated by the velocity field of a random dipole at the origin and numerically calculate the fractal dimension of the particle trajectory after a finite number N of time steps of the fixed size $\delta t$. The authors claim that the fractal dimension $D_f$ of this trajectory as a $D_f$-dimensional object immersed in the D-dimensional space is smaller than D and weakly depends on the system parameters: $N$, $\delta t$, $L_f$.

I) First of all, one can see that there are many references to the companion paper [23] (as well as to [24] in Sec. 6) which makes the current numerical work not standalone and reduces the possibility to assess it independently. I strongly recommend to merge two companion submissions into the one.

II) Second, the relevance of the model considered by the authors to the turbulence is doubtful, especially taking into account the suggested boundary conditions 1 and 2. Indeed, following the discussion in Sec. 6.1 about the periodically located dipoles that should model the vortices in the periodically boundary condition 1, one can immediately see that - All these "dipoles" are considered to be the copies of the same dipole with the completely correlated dipole moments for all of them. - The dipolar fields are cut at the distance $L_f$ from each dipole and do not penetrate to the next elementary cell. This makes all these dipoles to be short-range.

In addition, in the introduction the relevance of the considered model with the "velocity field caused by a “dipole” with a randomly modulated moment located at the origin" should be motivated by the references in the literature. For a moment, reading the introduction, one cannot understand how the only dipole at the origin of the infinite system can affect the dynamics in 3D system where the probability of the return to this point is zero in the thermodynamic limit. Please clarify all this in the introduction from the very beginning.

III) The boundary condition 2 does not have any proper physical motivation from the turbulence. It is just the calculation of the fractal dimension of the object formed by the overlap of several independent particle trajectories started at the same initial point and escaped beyond the boundary of the D-dimensional cube. Please clarify the relevance of this boundary condition 2 to the physical system.

IV) The authors consider a D-dimensional torus for the periodic boundary conditions. I wonder what will happen at the surfaces with different topology (genus): D-dimensional sphere of genus>1 surfaces?

V) In addition to the boundary conditions, one should carefully consider the definition of the fractal dimension, Eq. (8), and its dependence on the parameters. The definition (8) of $D_f$ is given by the fractal dimension of the trajectory as a $D_f$-dimensional object immersed in the D-dimensional space. Unlike any static objects in an infinite D-dimensional space, the number of occupied boxes by a particle trajectory changes with the number of time steps $N$, with the system size $L_f$, with the boundary conditions and even with the time step $\delta t$. As a result, the definition of the fractal dimension is highly non-universal and system-specific.

VI) Consequently, by using the above definition of $D_f$ the authors have to consider the parameter dependence of $D_f$, which they partially did. Here I would like to ask several questions: 1. What is the reason to consider the concrete range of parameters (9-11)? How do the results change in the other range? What do the overlapping regions for $d_h$ mean in (9)? 2. How these range of parameters are related to the results in Tables 1-4? 3. What are the values of all the parameters ($N$, $L_f$, $\delta L$, initial point) in Tables 1-4? In the current order, the results of the Tables 1-4 are unclear for a reader as they are given before the ones from Sec. 4. Please consider to move the tables later. 4. Why $D_f=0$ and $\sigma = 0$ for $d_h>0.2$ in Table 3? It is unclear from the table. 5. The dependence of $D_f$ versus $\log_10(N)$ given in Fig. 3 does not show any saturation. From the general point of view $D_f$ should scale with $N$ as the increase of the time steps increases the number of the boxes occupied by a trajectory. Therefore the statement "When the parameter $N$ (the number of steps) varies ... the fractal dimension $D_f$ asymptotically approaches a constant" is at least misleading. 7. In the same way, $D_f$ depends on the system size as by decreasing $L_f$ (or increasing $N$) one can increase $D_f$ to $D$ for any dimensionality $D$. 8. Comparing Fig. 3 with the above statements, one cannot understand how the results for $D_f$ in Tables 1-4 are obtained. As $D_f$ cannot "asymptotically approach a constant, which is equal to the values given in the previous section", it is unclear - which finite values of $N$ and $L_f$ are used in Tables 1-4 and why, - what all this finite-size and finite-time values of $D_f$ have in common with any fractal dimension of Levy flights and anomalous diffusion (e.g. to the one considered in [23]). Please clarify the above statement of the manuscript about $D_f$ and the definition of the fractal dimension used for a finite system size and a finite number of time steps. 9. The dependence on the initial location $x_0$ of the particle is also quite unclear. Indeed, like in the item VIII) below, the entire Fig. 6 is relevant for the case of the missing particles of [23] (or partially to the boundary condition 2), but not related to the periodic boundary conditions. In this sense the phrase "the fractional dimension is not well-defined when the initial particle location is too close to the dipole" is misleading for the periodic boundary conditions 1. 10. The dependence of the critical position $r_c$ on $\delta t$ and $L_f$ in (13) shows the strong dependence of the results both on $\delta t$ and $L_f$ which is neglected by the authors. Please clarify this issue in the text. 11. The dependence of the results on the time step $\delta t$ and of the size of the counting box $\delta L$ is not considered by the authors.

VII) In addition, for solving continuous-time Eq. (6) the authors use the simple Euler integration scheme (7) with the fixed time step $\delta t = 0.01$. Why do not they use any Runge-Kutta integration scheme? Does the Euler scheme conserve some integrals of motion? What is the accuracy of Euler scheme? How the results depend on the choice of $\delta t = 0.01$?

VIII) Figure 3 and the discussion around it is unclear. Indeed, Eqs. (1-8) deal with a single particle, while the discussion of the section "Missing particles" includes "the number of particles". It seems that the authors consider many random realizations of the same stochastic process and count the fraction of particles which cross the border of the D-dimensional cube of the size $2L_f$. This is especially unclear with respect to the boundary conditions 1 and 2 which do not have any missing particles by definition. This part seems to be relevant for the companion manuscript [23] as there the fraction of missing particles in such random realizations is a relevant parameter of the probability loss. This once again shows that the current submission should be merged with the one of [23].

IX) In addition to it, the authors claim that the above number of missing particles decreases exponentially in Fig. 3 (with the time step, I guess) which is not the case in [23], where Fig. 2 shows the approximately logarithmic decay (linear with $\log_2(t)$). This issue should be clarified.

To sum up, I cannot recommend the manuscript for the publication in SciPost Physics in the current form. I may reconsider my decision if the authors address all my questions and comments, including the one about the merging with the companion submission (https://scipost.org/submissions/2201.04900v2/).

Requested changes

1 - Please merge this submission with [23] in order to make both projects consistent, standalone and results clear.

2 - Please consider to change the description of the model from too generic one in (1-3) to the concrete one in (4-6): The current description in (1) with randomly moving source locations $\zeta_i$ are confusing for a reader. The only relevant parameter $d(t)$ in (5-6) improves the clarity and the readability of the manuscript.

3 - Please reconsider and clarify the definition and the parameter dependence of the fractal dimension (see the report).

4 - Please address all other questions and suggestions from the report.

Minor changes: 1) - Several notations are not defined, defined incorrectly, or defined too late in the text. Please add the proper definition in words in the corresponding places: - After Eq. (1) please replace "is the location of the source" by "are the locations of the sources". - In the footnote 1: please define SLE (used in [23]). - After Eq. (3) please define the relations between $\zeta_i$ at different time steps: are they considered to be completely uncorrelated and random? If yes, why this choice is physically relevant? - $d_h$ is not defined in Eq. (6), only in the discussion after Eq. (7). - double usage of the notation "n" in Eq. (1) for the number of sources and in Eq. (8) for the number of boxes occupied by the particle trajectory. - Subsection "Missing particles" in page 8: the notion of "the number of particles" is not defined. Eqs. (1-8) are written for the only particle: please clarify the definition of the number of particles (probably with the number of numerical realizations). - In the same subsection and page the scalar $x_0$ of the initial condition in the D-dimensional space is vaguely defined. What does this scalar mean as a coordinate in the D-dimensional space? - Eq. (14) in page 11: $\omega$ and $\Gamma$ are not defined, as well as the subscripts $_{1,2,3}$. - Double usage of the notation "N" in Sec. 3 as a number of time steps and after (17) as a number of dipoles. - The abbreviation "N-S" after Eq. (20) is not defined. - "The energy dissipation rate per unit mass" in Sec. 6.3 is not properly defined. It is not clear: "per unit mass" of what? Eqs. (1-7) are written for the only particle.

2) - Please add axis labels to figures 1 and 2.

3) - Please add proper descriptive captions to all figures in order to avoid searching through the text of their discussion.

4) - Please make figures in vector-graphic format to avoid degradation of its quality in a raster format.

5) - The entire section 6 "Discussion" is put after the summary and all the results, therefore its relevance is doubtful. If the authors consider this section to be important, they should move it to the relevant place of the manuscript.

  • validity: low
  • significance: low
  • originality: good
  • clarity: ok
  • formatting: acceptable
  • grammar: reasonable

Author:  Yoshiki Matsuoka  on 2022-05-03  [id 2432]

(in reply to Report 2 on 2022-03-09)

We have resubmitted the paper and have attached our reply in the attached pdf.
Thank you for your cooperation.

Attachment:

reply_JqXdeax.pdf

---

## Editorial Decision

awaiting_resubmission